# Spatio-Temporal Dynamics of Terminal Lakes in the Hexi Interior, China

Qin Ma [1,2], Xiaojun Yao [1,2,3,*], Cong Zhang [1,2], Chen Yang [1,2], Kang Yang [1,2], Zhijuan Tian [1,2] and Jiawei Li [1,2]

1   College of Geography and Environmental Science, Northwest Normal University, Lanzhou 730070, China; 2021222915@nwnu.edu.cn (Q.M.); 2021120206@nwnu.edu.cn (C.Z.)
2   Key Laboratory of Resource Environment and Sustainable Development of Oasis, Lanzhou 730070, China
3   Academician's Studio of Gansu Dayu Jiuzhou Space Information Technology Company Limited, Lanzhou 730050, China
*   Correspondence: xj_yao@nwnu.edu.cn; Tel.: +86-131-0930-9241

**Abstract:** The evolution of a terminal lake at the end of a river not only reflects the climate change characteristics within the basin but also the impact of regional human activities, especially in arid areas. In the Hexi Interior of China, three terminal lakes (e.g., Halaqi Lake, East Juyanhai Lake, and Qingtu Lake) situated in the Shule River, Heihe River and Shiyang River, respectively, have been increasingly studied to support regional ecological protection and sustainable oasis development. In this study, Landsat TM/ETM+/OLI and Sentinel-2 MSI imagery were used to examine Halaqi Lake spanning from 2017 to 2022, East Juyanhai Lake from 1990 to 2022, and Qingtu Lake from 2009 to 2022. The focus of this investigation was to characterize changes in lake area and the impact of climate change and human activities. The results revealed a dramatic change in Halaqi Lake, which suddenly emerged in 2017, initially covering an area of 13.49 km$^2$, gradually vanishing nearly in 2021, and reappearing in 2022 with a reduced area of 9.53 km$^2$. The area of East Juyanhai Lake was 54.39 km$^2$ in 1990 but reduced to 40.84 km$^2$ by 2022. Throughout this period, it encountered episodes of drying up in 1992, 1995, 2001, and 2002. Qingtu Lake emerged in 2009, with an area of 0.09 km$^2$, and subsequently expanded to 2.60 km$^2$ by 2022. Climate change and human activities collectively influence the area fluctuations of these three terminal lakes. Among these factors, temperature changes have a greater impact on the lake area in East Juyanhai. Global warming has worsened glacier melting in the Qilian Mountains, resulting in increased inflow in certain years and substantial lake area expansion. Human activities are the primary drivers of changes in Halaqi Lake and Qingtu Lake. Industrial water consumption is the key factor influencing area changes in Halaqi Lake, whereas water usage in forestry, animal husbandry, and fisheries plays a dominant role in the area changes of Qingtu Lake. Furthermore, the introduction of ecological water conveyance projects has had an indispensable effect on rejuvenating and preserving the watershed areas of these three terminal lakes. It is important to emphasize that human-driven water resource management is the primary cause of sudden changes in the lake areas.

**Keywords:** terminal lake; climate change; ecological water conveyance; Hexi Interior

## 1. Introduction

Terminal lakes are crucial to the terrestrial water cycle, serve as pronounced indicators of climate change, and show high susceptibility to anthropogenic environmental shifts [1]. Terminal lakes often reside in interior areas far from the ocean and typically exist within arid climates with underdeveloped water systems [2]. These lakes form at the mouths of rivers that cannot flow out, earning names such as "terminal lakes", "river mouth lakes", or "endorheic lakes." Belonging to the categories of inland and non-draining lakes, these bodies of water are predominantly saline due to their low elevations, extremely arid climate, limited rainfall, and high evaporation rates. River runoff serves as the primary water source for these lakes, with short intermittent streams flowing into them and providing limited replenishment [3]. Terminal lakes heavily rely on upstream mountain regions for water sources and locate

themselves at the very end of the river basin. They are profoundly influenced by human activities in the middle and lower reaches of the basin [4]. Nevertheless, terminal lakes also demonstrate remarkable sensitivity to climate change. The shaping of these lakes is driven by the combined effects of climatic shifts and human interventions [5], with the latter often inducing significant hydrological pattern alterations [6]. Many interior basins shift from a natural water cycle to one heavily influenced by human activities [7]. Particularly in arid zones with ecologically vulnerable lakes, human activities have a pronounced impact [8]. Activities like land reclamation [9], dam construction [10], urban water consumption [11], and mineral extraction [12] significantly influence lake dynamics. These actions influence lake reduction rates, salinization, and lead to challenges such as habitat degradation and the emergence of sand and dust storms [13]. Regional authorities initiate conservation measures and water management strategies in response to the ecological threats from these developments [14].

The Hexi Interior is central to China's Silk Road Economic Belt and plays a vital role in water sourcing for the arid northwestern region [15]. Terminal lakes in this region play a critical role in ecological balance and biodiversity preservation. Halaqi Lake, East Juyanhai Lake, and Qingtu Lake receive water from the Shule, Heihe, and Shiyang Rivers, respectively, which originate in the Qilian Mountains. The glacial and snowmelt water originating from the Qilian Mountains serves as a crucial source of replenishment for rivers in the area [6]. These rivers supply water that benefits urban areas, agriculture, and reservoirs while traversing diverse channels [16]. Yet, the over-allocation of surface water reduces inflow to the lakes, which compromises the lakes' essential ecological functions, such as windbreak, sand stabilization, and biodiversity sustenance [11]. The ecological significance of the terminal lakes in the Hexi Interior underscores the importance of understanding their dynamics and the external factors influencing them to ensure effective conservation.

To elucidate the hydrological evolution of interior basins in the Northwest Arid Zone, some studies have been conducted to examine changes in water resources within the basins using remote-sensing image data. These studies also analyze the influence of meteorological data and human activities on the water balance. As an illustration, it was observed that from 1960 to 2018, climate change played a predominant role in shaping the runoff in the upper Heihe River, while human activities had a relatively minor impact. In contrast, the runoff in the mid-river region was less influenced by climate change, with human activities emerging as the dominant factor [17]. While the ecological water conveyance project in the Heihe River Basin has aided the expansion of the East Juyanhai Lake, challenges persist. The average groundwater level in the basin's middle reaches keeps declining, water distribution disparity between the middle and lower reaches grows, and the program falls short of ensuring ecological restoration and sustainable development across the basin [18]. The runoff of the Shiyang River witnessed a decline from 1956 to 2009. The fluctuations in the Caiqi Hydrological Station runoff before 1968 were primarily attributable to climate change. However, the alterations in Caiqi Hydrological Station runoff after 1968 were the outcome of the combined influence of both climate and land-use changes. Importantly, it was observed that land-use change had a considerably larger impact on runoff compared to climate change during this period [19]. After the ecological water conveyance, the groundwater depth in Qingtu Lake at the Shiyang River's terminal increased, progressively extending from the water's edge to the desert boundary. Additionally, there was a notable augmentation in the overall vegetation cover throughout the region [20].

Halaqi Lake, East Juyanhai Lake, and Qingtu Lake, as terminal lakes in arid regions, all possess unique water sources and ecological environments. These lakes rely not only on river surface runoff and atmospheric precipitation but also on glacial meltwater from the Qilian Mountains to maintain their water levels. Simultaneously, the hydrology and ecosystems of these lakes are significantly influenced by upstream oasis cities. The use and management of water resources in oasis cities directly affect the quantity and quality of water in the lakes, thereby impacting the ecological health and stability of the lakes. Moreover, the changing trends and driving factors of these three lakes generally show similarities. While previous studies concentrated on changes in individual terminal lakes, an overall analysis of changes in

multiple terminal lakes in the region and the influencing factors remains lacking. This study employs a quantitative approach to examine changes in terminal lake areas using diverse remote-sensing data. It also investigates the factors behind these changes by integrating meteorological, hydrological, and human activity data. The ultimate goal is to furnish a solid scientific foundation for water resource allocation and policymaking in the region.

## 2. Study Area

The Hexi Interior Basin (93° E–104° E, 37° N–42° N) lies in the northwest of Gansu Province and extends from the Wushao Ridge in the east to the southern regions of the Qilian Mountains and the Altun Mountains watershed. It borders the Xinjiang Uygur Autonomous Region to the west and reaches northward to the border of the Inner Mongolia Autonomous Region and Mongolia [21] (Figure 1). From west to east, the distribution encompasses three major water systems: the Shule River, Heihe River, and Shiyang River. All these systems originate in the Qilian Mountains, with glacier snowmelt acting as the primary source of recharge. The region in a temperate continental arid climate zone has a gradient of decreasing precipitation from east to west, high potential evaporation rates, and pronounced temperature variations [22]. The three primary water systems undergo consumption via infiltration and irrigation and converge into terminal lakes in depressions. These lakes are identified as Halaqi Lake, East Juyanhai Lake, and Qingtu Lake. Halaqi Lake (92°52′ E–92°54′ E, 40°17′ N–40°22′ N), at the terminal of the Shule River, marks the intersection of the western boundary of the Dunhuang West Lake National Nature Reserve and the eastern edge of the Kumtag Desert [23]. East Juyanhai Lake (101°11′ E–101°19′ E, 42°15′ N–42°20′ N), also known as Subo Naoer, is the terminal of the Heihe River and lies in the Alxa League's Ejin Banner. Historically, Juyanhai Lake was one of the largest lakes in the Northwestern Region [24]. Qingtu Lake (103°34′ E–103°39′ E, 39°04′ N–39°10′ N) is situated at the conclusion of the Shiyang River. In antiquity, it was part of the Xiutu Lake group of lakes and boasted a water area second in size only to Qinghai Lake [25]. Environmental transformations and human activities have led to successive size reductions and even complete desiccation of the three terminal lakes. This has given rise to progressively severe ecological issues, including land cover degradation, frequent occurrences of sandstorms, and prolonged periods of drought. Recent climate patterns with warming and increased humidity, along with human interventions, have been pivotal in preventing further deterioration of the lake environments.

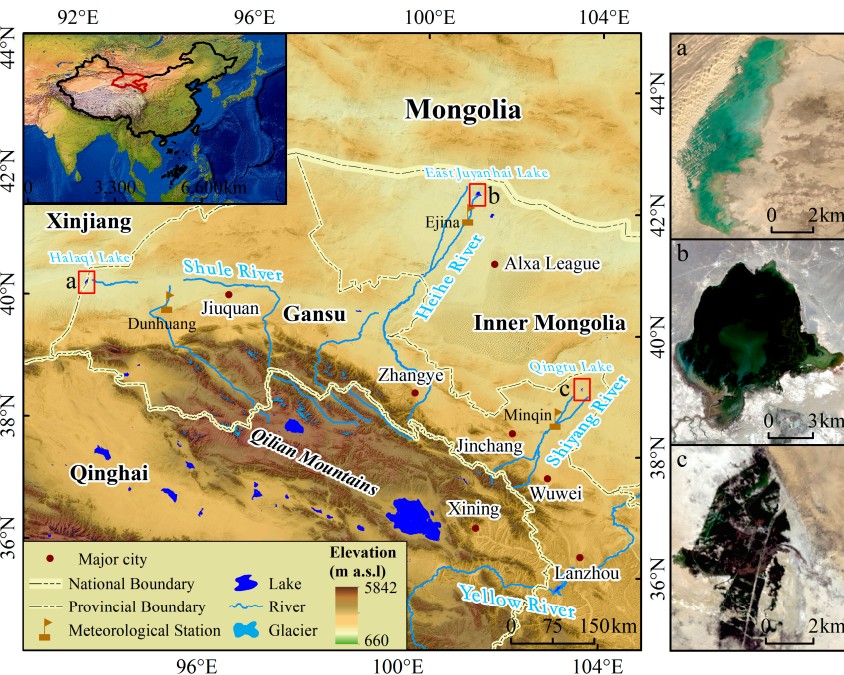

**Figure 1.** Geographic locations of Halaqi Lake, East Juyanhai Lake, and Qingtu Lake.

## 3. Data and Methods

### 3.1. Data Sources

#### 3.1.1. Landsat and Sentinel Series Images

This research utilized Landsat TM/ETM+/OLI images and Sentinel-2 MSI images from the Google Earth Engine platform (https://earthengine.google.com, accessed on 24 July 2023) that span from January to December and cover the years 1990 to 2022. These images were used for the purpose of delineating the lake boundaries. Between 1990 and 2022, a total of 819 images were downloaded. The specific distribution of Landsat TM/ETM+/OLI and Sentinel-2 MSI images acquired during this period is illustrated in Figure 2. Among them, there were 237 scenes of Landsat TM images from 1990 to 2011, 28 scenes of Landsat ETM+ images in 2012, 342 scenes of Landsat OLI images from 2013 to 2022, and 212 scenes of Sentinel-2 MSI images from 2015 to 2021. The images for each year were acquired in a month-by-month order to obtain lake shoreline information. In cases where data for specific months were missing, the closest available month within the same year was used as a substitute. Remote-sensing images with the least cloud cover were selected for optimal quality. All imagery obtained through the Google Earth Engine (GEE) platform undergoes a comprehensive pre-processing pipeline, which includes radiometric, geometric, topographic, and atmospheric corrections. The Sentinel-2 MSI utilizes a spatial resolution of 10 m for its selected bands, while Landsat TM has a spatial resolution of 30 m for its selected bands. To enhance the spatial resolution of the images while preserving their multispectral information, band fusion is applied to the visible and panchromatic bands of Landsat ETM+ and OLI images to achieve a spatial resolution of 15 m.

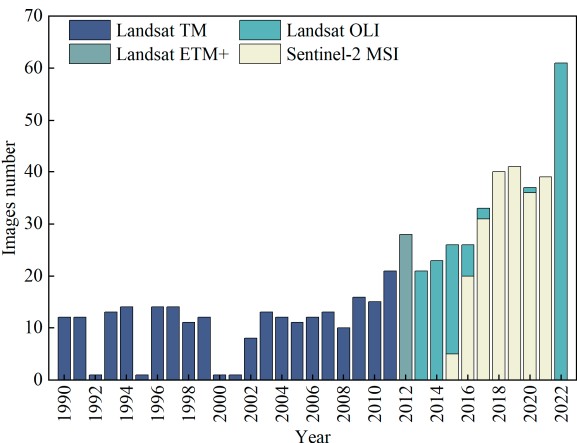

**Figure 2.** Count of Landsat TM/ETM+/OLI and Sentinel-2 MSI images.

#### 3.1.2. Meteorological Data

Temperature, evapotranspiration, and precipitation are the three main meteorological factors affecting the water volume of lakes. In lakes recharged by glacial meltwater, warmer temperatures can have both a positive effect by increasing the recharge of glacial meltwater to lakes and a negative effect, due to enhanced evapotranspiration. Evapotranspiration is inversely positively proportional to the water volume of the lakes, while precipitation is proportional to it. To analyze the effects of climatic factors on Halaqi Lake, East Juyanhai Lake, and Qingtu Lake, this study used monthly surface climate element data from Dunhuang meteorological station (2017–2022), Ejina Banner meteorological station (1990–2022), and Minqin meteorological station (2009–2022) provided by the National Meteorological and Scientific Data Sharing Service Platform (http://data.cma.cn, accessed on 18 August 2023). The study analyzed the inter-annual trends of mean temperature and precipitation and calculated potential evapotranspiration to explore the responses of the lake areas to climate change in a regional climate context. The Dunhuang meteorological station is situated in the southeast of Halaqi Lake, approximately 150 km away from the lake. The Ejina and Minqin meteorological stations are located in the southwestern part of East

Juyanhai Lake and Qingtu Lake, respectively, at distances of approximately 40 km and 70 km from their respective lakes. These meteorological stations effectively represent the climatic characteristics of the study area.

### 3.1.3. Water Resource Consumption Data

For terminal lakes that rely on upstream river recharge, the amount of water withdrawn by the river is an important factor affecting the amount of water in the lake [26]. Amidst severe water resource constraints affecting socio-economic development in the region, the amount of water used for human production and domestic use has a direct impact on the amount of water withdrawn from the rivers and, consequently, on the amount of water in the lakes [27,28]. Water for human production is primarily allocated to agriculture, industry, forestry, animal husbandry, and fisheries. Domestic water usage encompasses residential and urban public water, and it also includes ecological water. In the Hexi Interior Basin, the increasing utilization of water for production and living predominantly relies on rivers and groundwater. The substantial reduction in replenishment from the three major inland rivers to terminal lakes has consequently affected lake areas. Presently, the Hexi region grapples with a severe water resource crisis. This crisis intensifies the conflict between diminishing lake water resources and rising human water demand. Consequently, achieving a balance in the allocation of water resources between rivers and lakes has become a vital theme in the region's water resource management. The volume of water utilized for human production and living serves as a foundational dataset for studies on lake area changes and the dynamic equilibrium of regional water resources. It provides crucial data support for the rational allocation of regional water resources. The water resource consumption data includes water consumption for various purposes such as agricultural irrigation, industrial, residential living, forestry, animal husbandry, and fisheries, urban public, ecological environment in the Shule River Basin (2017–2021), Heihe River Basin (1994–2021), and Shiyang River Basin (2009–2021), as well as the ecological water conveyance into the three terminal lakes. Building on previous relevant research [29,30], this study utilized ecological water conveyance data. Data sources included the Danghe Reservoir for Halaqi Lake; Yingluo Gorge Hydrological Station, Zhengyi Gorge Hydrological Station, and Langxinshan Hydrological Station for the East Juyanhai Lake; and the Caiqi Hydrological Station and Hongyashan Reservoir for Qingtu Lake. Water consumption data in the Shule River Basin, Heihe River Basin, and Shiyang River Basin were also used to analyze interannual variations in water consumption and ecological water discharge resulting from human activities. This analysis aims to elucidate the impact of human activities on changes in the area of terminal lakes. Water resource consumption data were sourced from the Gansu Water Resources Bulletin (http://slt.gansu.gov.cn, accessed on 24 August 2023).

### 3.2. Method

### 3.2.1. Extraction of Lake Information

The Normalized Difference Water Index (NDWI) is widely employed as the primary method for delineating the boundaries of water bodies [31]. This study used NDWI for the extraction of water bodies in the terminal lakes. A triangulation method for image binarization determined an appropriate threshold value to extract water body information. The calculation formula for NDWI is as follows:

$$NDWI = \frac{\rho_{Green} - \rho_{NIR}}{\rho_{Green} + \rho_{NIR}} \tag{1}$$

where $\rho_{Green}$ and $\rho_{NIR}$ represent the reflectance in the green and near-infrared bands, respectively.

Zack, G.W. [32] introduced an adaptive threshold selection method for image binarization. This method aims to determine the optimal threshold based on the histogram distribution of the image. When the maximum peak of the histogram is closer to the brightest side, the gray level that maximizes the distance between the base of the triangle

and the histogram is selected as the optimal threshold. The formula for the triangular thresholding method is as follows:

$$(y - y_0) = \frac{y_1 - y_0}{x_1 - x_0}(x - x_0) \tag{2}$$

$$d = \left| \frac{Ax + By + C}{\sqrt{A^2 + B^2}} \right| \tag{3}$$

$$A = (y_1 - y_0) \tag{4}$$

$$B = -(x_1 - x_0) \tag{5}$$

where $x_0$ is the grayscale value at the valley bottom, and $y_0$ is the frequency of the histogram corresponding to the grayscale value at the valley bottom. $x_1$ is the grayscale value at the peak, and $y_1$ is the frequency of the histogram corresponding to the grayscale value at the peak. The variable $d$ is used to denote the vertical distance between a specific gray level and the base of the triangle.

This study extracted the water areas of Halaqi Lake, East Juyanhai Lake, and Qingtu Lake. The accuracy of the triangulation method and Otsu's method as two adaptive thresholding methods were compared for this extraction. The findings indicated that the areas extracted using the triangulation thresholding method demonstrated higher accuracy for Halaqi Lake, East Juyanhai Lake, and Qingtu Lake. The alignment of these extracted water bodies with visually interpreted results produced kappa coefficients of 0.92, 0.92, and 0.81, respectively.

3.2.2. Calculation of Potential Evapotranspiration

Understanding the atmospheric capacity to lose water is pivotal for lake water balance studies, especially for closed systems like Halaqi Lake, East Juyanhai Lake, and Qingtu Lake, where evaporation is the primary water loss mechanism. Potential evapotranspiration [33] is a measure that denotes the total volume of pure water evaporated from the earth's surface. It is influenced by factors such as solar radiation, air temperature, relative humidity, and wind speed. This study employed the Food and Agriculture Organization of the United Nations (FAO) modified Penman–Monteith equation. This equation, which requires inputs like temperature (average, maximum, minimum), humidity, sunshine duration, and wind speed, offers a comprehensive approach to quantifying potential evapotranspiration from lakes. The potential evapotranspiration formula is as follows:

$$ET_0 = \frac{0.408\Delta(R_n - G) + \gamma \frac{900}{T+273}u_2(e_s - e_a)}{\Delta + \gamma(1 + 0.34u_2)} \tag{6}$$

where $ET_0$ represents potential evapotranspiration (mm), $R_n$ stands for net radiation [MJ/(m²·d)], $G$ represents soil heat flux [MJ/(m²·d)], $\gamma$ is the psychrometric constant, $T$ is the daily average temperature (°C), $u_2$ is the 2-m wind speed (m/s), $e_s$ and $e_a$ represent saturated vapor pressure and actual vapor pressure, respectively (kPa), and $\Delta$ represents the slope of the saturated vapor pressure curve (kPa/°C).

## 4. Results

### 4.1. Inter-Annual Change

Halaqi Lake, East Juyanhai Lake, and Qingtu Lake do not demonstrate stability and instead display substantial fluctuations over time. These three lakes exhibited notable variations in both their lake area and shoreline length (Figure 3). Throughout the 32-year timeframe, all three terminal lakes underwent rapid expansion subsequent to periods of drying. However, all these lakes began showing a reduction trend in size starting from 2019. From 1990 (54.39 km²) to 2022 (52.97 km²), the combined area of the three terminal lakes has decreased by 2.61%. During this period, from 1990 to 2008, Halaqi Lake and Qingtu Lake remained continuously dry until November 2009 when Qingtu Lake formed a

water area of 0.09 km². Halaqi Lake shifted from its stable dry status in September 2017 to have a water surface area of 13.49 km².

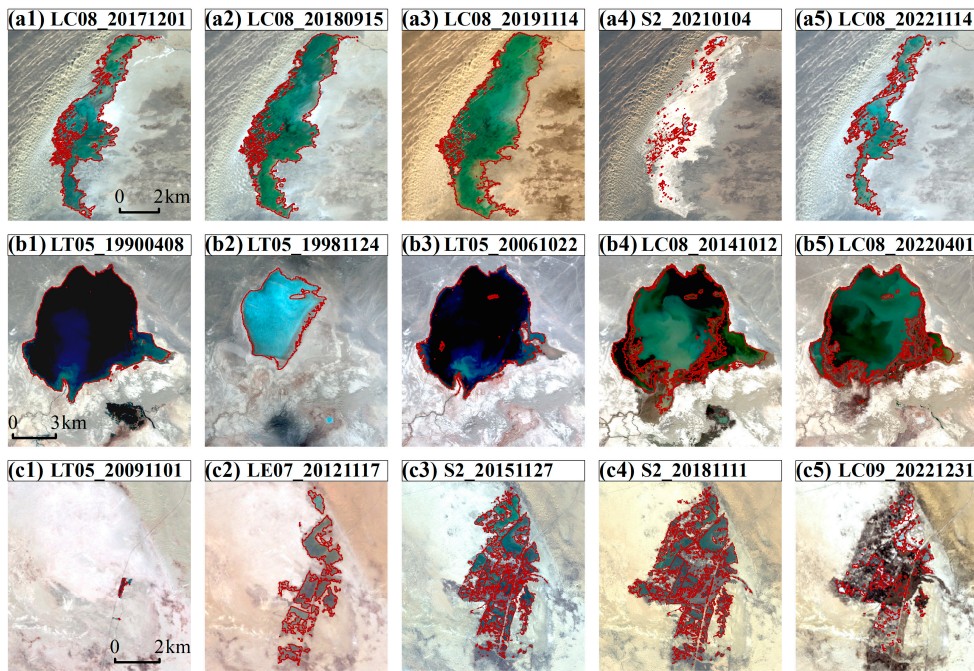

**Figure 3.** Evolutionary process of the shorelines of Halaqi Lake (**a1–a5**); East Juyanhai Lake (**b1–b5**); and Qingtu Lake (**c1–c5**).

Halaqi Lake experienced significant size and shoreline length changes between 1990 and 2022. The lake faced a drought from 1990 to 2016 (Figure 4(a1)). It then expanded rapidly from 2017 (13.49 km²) to 2020 (22.65 km²), peaking in size in 2019, at 23.03 km². The year 2021 brought a drastic size reduction to 1.44 km², marking a 93.75% decline from its 2019 size. The lake size recovered in 2022, expanding to 9.53 km² for an increase of 35.14%. Regarding shoreline length, Halaqi Lake expanded rapidly in 2017 to 193.44 km (Figure 4(b1)). Stability followed from 2017 to 2020, with a length of around 214.24 km. A sharp reduction to 81.9 km occurred in 2021, a decrease of 61.77%. The shoreline length recovered in 2022 to 115.08 km, reflecting a 15.49% increase.

In 1990, among the three terminal lakes, only East Juyanhai had water, and it covered an area of 54.39 km² (Figure 4(a2)). A significant shrinkage occurred from 1990 to 1991 (31.72 km²), marking a 41.68% decrease. The lake dried up in 1992, 1995, 2000, and 2001. A growth phase spanned from 2002 (25.07 km²) to 2020 (67.72 km²), with the lake's area increasing by 170.12%. Its largest extent was in 2019, at 69.10 km². A contraction phase followed in 2021 (45.82 km²) and 2022 (40.84 km²), resulting in a 39.70% size reduction from 2020. Overall, the lake's area trend leans towards growth. The shoreline of East Juyanhai Lake expanded over the years, peaking in 2018 at 721.22 km (Figure 4(b2)), a 190.91% increase from 1990 (247.92 km). Post-2018, the shoreline consistently contracted, with a significant 70.97% reduction between 2020 (675.76 km) and 2022 (196.2 km).

Qingtu Lake and Halaqi Lake have shown similar trends in area alterations. Qingtu Lake's area expanded 27-fold from 2009 (0.09 km²) to 2022 (2.60 km²), and the overall trend continues to be upward (Figure 4(a3)). From 1990 to 2008, Qingtu Lake experienced desiccation. A rapid expansion phase followed from 2009 to 2014 (9.69 km²). Fluctuations in the area occurred between 2015 (9.5 km²) and 2018 (9.83 km²), with the highest value in 2017, at 11.63 km². A contraction phase followed post-2018, with a 56.73% decrease between 2021 (6.02 km²) and 2022. The shoreline of Qingtu Lake generally expanded, peaking in 2018 at 387.42 km (Figure 4(b3)), a 105-fold increase from 2009 (3.66 km). A contraction began in 2019 (357.78 km), with a significant 67.96% reduction in shoreline length between 2021 (293.86 km) and 2022 (94.14 km).

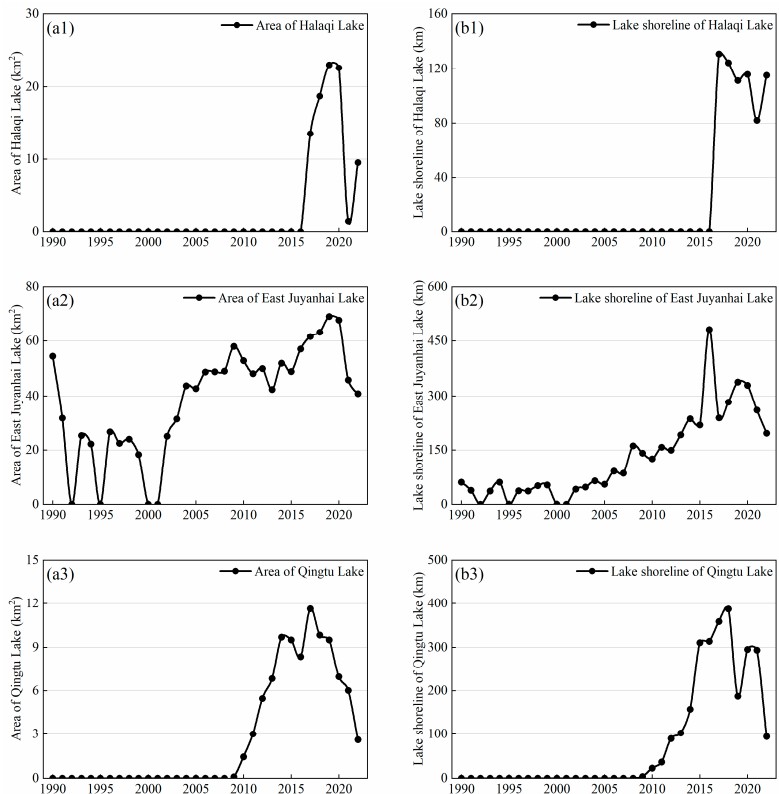

**Figure 4.** Changes in the maximum lake area (**a1**–**a3**); and maximum shoreline length (**b1**–**b3**) of Halaqi Lake, East Juyanhai Lake, and Qingtu Lake from 1990 to 2022.

### 4.2. Intra-Annual Change

The images of Halaqi Lake (2017–2022), East Juyanhai Lake (1990–2022), and Qingtu Lake (2009–2022) were divided into four phases to account for seasonal variations: February to April, May to July, August to October, and November to the subsequent January. For each phase, average values of the lake area and shoreline length were determined, with maximum and minimum values highlighted. Qingtu Lake showed the most pronounced intra-annual area fluctuations. In contrast, East Juyanhai Lake displayed the least area fluctuations but the most significant shoreline length variations. The May to July phase stood out for its area stability across all three terminal lakes (Figure 5).

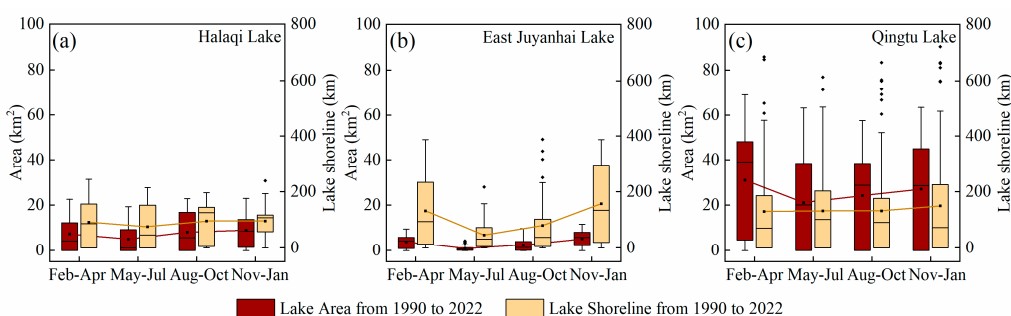

**Figure 5.** Changes in the area and shoreline length of Halaqi Lake (2017–2022) (**a**); East Juyanhai Lake (1990–2022) (**b**); and Qingtu Lake (2009–2022) (**c**). Each box plot presents the 25th, 50th, and 75th percentiles. Vertical whiskers denote the minimum and maximum values excluding outliers, and small squares represent the mean.

Halaqi Lake's shoreline length experienced its most significant fluctuations from February to April, and its length peaked during this phase. The May to July phase sees both the average lake area and shoreline length comparatively smaller, with the least area

fluctuations. In contrast, August to October has the most substantial area fluctuations. From November to the subsequent January, the lake area expands to its largest size, but the shoreline length remains stable. Notably, the shoreline length and lake area changes do not correlate positively. This discrepancy might arise from counting numerous small, scattered water bodies in the total shoreline length, causing inconsistent shoreline fluctuations relative to lake area changes.

East Juyanhai Lake displays the most stable lake area, especially when compared to Halaqi Lake and Qingtu Lake. Among these, the period from May to July is characterized by the highest level of lake area stability, with the smallest average area observed during this time. The values of 69.09 km$^2$, 67.72 km$^2$, and 62.33 km$^2$ represent significant outliers during this period. An abundance of water in East Juyanhai Lake from 2018 to 2020 accounts for these anomalies, leading to larger lake areas compared to those observed in the same quarter of other years. From November to the following January, the lake area undergoes the most substantial fluctuations and represents the period with the largest lake area. Notably, in East Juyanhai Lake, changes in shoreline length exhibit a positive correlation with changes in lake area.

Qingtu Lake experiences the most pronounced fluctuations in both area and shoreline length among the three terminal lakes. The period from February to April represents the time when the lake area reaches its maximum value within the study period, and it also has the highest average lake area. In contrast, during the May to July period, Qingtu Lake exhibits the smallest average lake area, yet it concurrently boasts the longest shoreline length among the observed phases. During the period from November to the following January, both the lake area and shoreline changes exhibit the most significant magnitude among the four phases, and the average shoreline length is the largest.

## 5. Discussion

### 5.1. Response of Lakes to Climate Change

Temperature, evaporation, and precipitation are meteorological parameters that exert influence on alterations in lake surface areas. Nevertheless, distinct lakes exhibit considerable variability in their responsiveness to climate fluctuations [34]. Within the Third Pole region, lakes adjacent to glaciers exhibit a more pronounced expansion trend and larger area changes when contrasted with non-glacier-fed lakes [35]. In Antarctica's McMurdo Dry Valleys, glaciers continuously provide water sources to the terminal lakes at the valley bottoms after melting, and some terminal lakes formed at the ends of glaciers are continuously expanding [36]. Global warming is causing the melting of glaciers and high-altitude snow, thereby accelerating the expansion of lakes [37]. The correlation between the surface area of East Juyanhai Lake (1990–2022) and Qingtu Lake (2009–2022) with temperature changes is strong (both > 0.60), while it is weaker for Halaqi Lake (2017–2022) (Figure 6). The distinct geographical locations of the three lakes account for the differences, as these lead to variations in water supply sources. All three lakes fall within the Qilian Mountains glacier snowmelt recharge region. Yet, the effects of global warming vary substantially across regions. The Halaqi Lake region recorded minimal temperature changes from 2017 to 2022, with a rate of 0.01 °C/a (Figure 7(a1)). In contrast, the East Juyanhai Lake region from 1990 to 2022 and the Qingtu Lake region from 2009 to 2022 experienced more pronounced annual average temperature change rates of 0.04 °C/a, and 0.09 °C/a, respectively. Glaciers are a vital freshwater resource in the interior of the northern Qilian Mountains region, with particularly strong resource effects [38]. Studies indicate a decline of 20.88% in the glacier area and 20.26% in ice storage in the Qilian Mountains from 1956 to 2010 [39]. This notable reduction in glacier area mainly results from the rapid shrinkage of small glaciers. Notable temperature changes directly result in accelerated glacier melt and snowmelt, consequently leading to an increase in water supply sources. This stands as the primary cause behind the conspicuous expansion of the lakes.

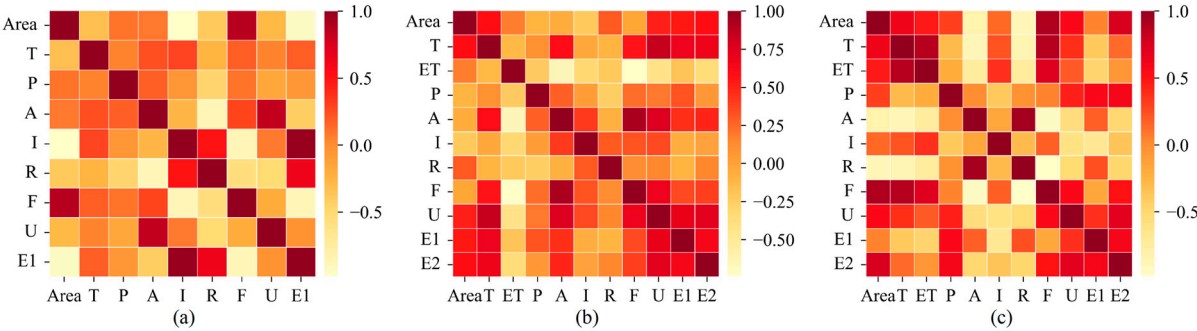

**Figure 6.** Correlations between the area of Halaqi Lake (**a**); East Juyanhai Lake (**b**) Qingtu Lake (**c**). Climate factors (T for temperature, ET for potential evapotranspiration, P for precipitation), watershed water usage (A for agricultural irrigation, I for industrial, R for residential living, F for forestry, animal husbandry, and fisheries, U for urban public, E1 for ecological environment), and ecological water conveyance (E2).

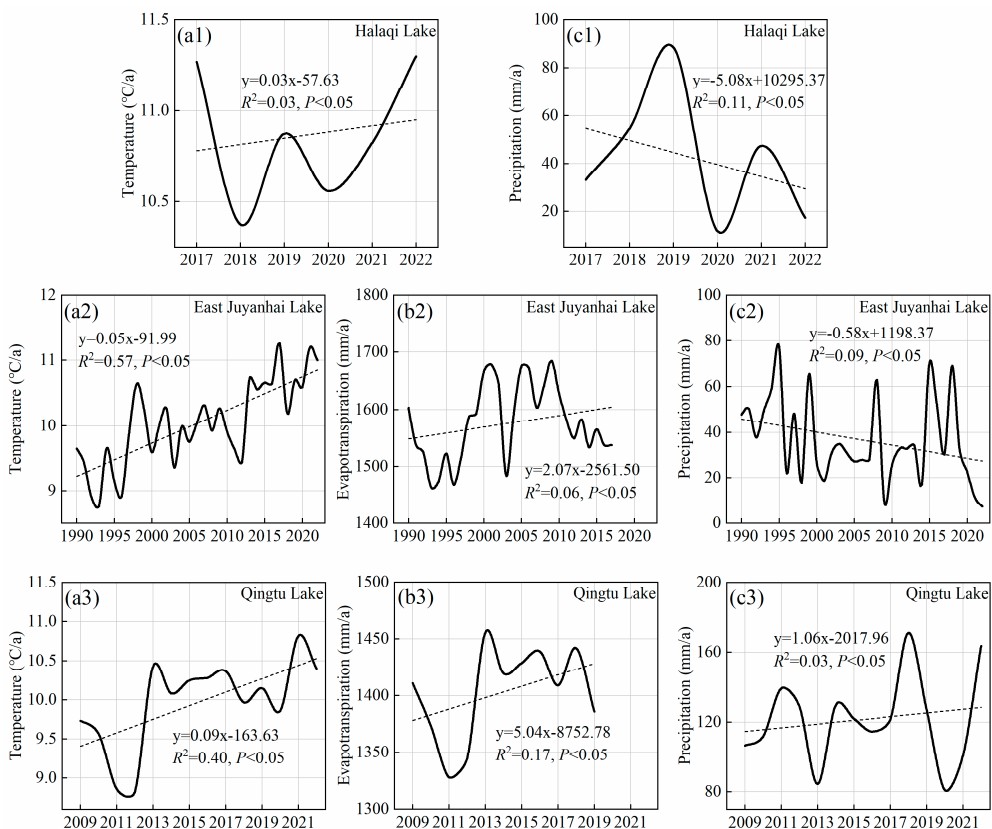

**Figure 7.** Changes in temperature and precipitation in the Halaqi Lake region from 2017 to 2022 (**a1,c1**); temperature, potential evapotranspiration, and precipitation changes in the East Juyanhai Lake region from 1990 to 2022 (**a2,b2,c2**); and temperature, potential evapotranspiration, and precipitation changes in the Qingtu Lake region from 2009 to 2022 (**a3,b3,c3**).

　　　Evaporation is a primary factor contributing to water loss in terminal lakes. Insufficient precipitation or runoff to offset increased evaporation rates leads to a reduction in a lake's water level and surface water coverage [40]. The potential evaporation in the East Juyanhai Lake and Qingtu Lake regions is overall on the rise (Figure 7). With potential evaporation reaching around 1400 mm per year and significantly exceeding precipitation, the Hexi region exhibits a drought trend. It is noteworthy that there is no apparent correlation between potential evaporation and changes in the lake area. This result suggests that changes in the lake area do not necessarily exhibit sensitivity to individual climate factors.

Lake shrinkage, or drying, can arise as a result of reduced precipitation, while extreme precipitation events can directly alter lake water levels [41]. Halaqi Lake (2017–2022) and East Juyanhai Lake (1990–2022) show a relatively low correlation with precipitation changes. In contrast, Qingtu Lake's area (2009–2022) correlates more with precipitation changes. The Hexi region houses all three lakes and experiences precipitation variations due to terrain influence. The eastern and southern sectors receive higher precipitation, while the western and northern zones receive less [42]. Precipitation levels for both Halaqi Lake (2017–2022) and East Juyanhai Lake (1990–2022) exhibit a declining trend, with rates of change at $-3.16$ mm/a and $-1.27$ mm/a, respectively (Figure 7(c1,c2)). In the Hexi region, the flood season from May to October accounts for 89.2% of the total annual precipitation [43]. The 2017 flood season saw a notable rise in precipitation in the Shule River Basin, leading to several flood events in the Shule River that surpassed warning water levels and a 4% increase in rainfall from the previous year. This event had an extremely important impact on the reappearance of Halaqi Lake in 2017 and the formation of a water surface covering an area of 2.22 km$^2$. Conversely, East Juyanhai Lake's area negatively correlates with precipitation changes due to an 85% drop in annual precipitation over 32 years. Precipitation reached 77.30 mm in 1995 but dropped to 7.30 mm by 2022, exacerbating the region's drought and shrinking the lake area. Qingtu Lake, however, saw an increase in annual precipitation from 2009 to 2022, growing by 4.38 mm/a, or 35% in total. While 2018 had high precipitation at 171.10 mm, 2020 recorded the lowest amount of precipitation at 81 mm. Precipitation rose again to 163.7 mm in 2021–2022. This increase in precipitation contributed to the expansion of Qingtu Lake's water area to some extent.

In summary, area changes in Halaqi Lake, East Juyanhai Lake, and Qingtu Lake exhibit the most significant correlation with temperature changes—a consensus observed in all three lakes. However, area changes display different directions of correlation: Halaqi Lake shows a negative correlation between its area and temperature, while East Juyanhai Lake and Qingtu Lake exhibit a positive correlation between area and temperature. This divergence may stem from differing geographical locations, resulting in distinct influences of climatic factors on each lake's area.

Fluctuations in lake areas stem from the intricate interplay between climate change and human activities rather than a single causal factor [44–46]. From a climate change perspective, global warming resulting in increased temperatures and alterations in precipitation patterns directly impacts lake water volume and surface area. Elevated temperatures accelerate glacier melt augmenting downstream lake water sources. Concurrently, a warmer and more humid climate trend may spur heightened rainfall, directly fostering lake area expansion. Human activities also contribute significantly to lake area dynamics. Increased human activity intensifies the demand for water resources, leading to overexploitation and resource utilization. Moreover, direct human interventions in lakes and rivers, such as dam construction and water transfer projects, disrupt the natural state of water bodies, influencing natural lake area fluctuations. An intricate interaction exists between climate change and human activities. For instance, climate-change-induced droughts and reduced water resources may compel increased human engagement in water resource development and management. These activities, in turn, may exacerbate local climate changes, forming a complex feedback loop.

### 5.2. Impact of Human Activities on Lake Area

Human activities and socioeconomic developments have added complexity to lake water changes. Such complexities are revealed through an imbalance in lake water resource supply and demand [47]; disruptions in the native systems coupling lakes and rivers [48]; and rises in lake water levels due to ecological water conveyance policies [49,50]. The Hexi Interior relies mainly on water from three river basins: the Shule River, the Heihe River, and the Shiyang River [51]. These rivers and their tributaries play a pivotal role in supplying water to Halaqi Lake, East Juyanhai Lake, and Qingtu Lake [52,53]. As a result, the areas of these three terminal lakes are significantly influenced by human activities.

The correlation analysis (Figure 6) reveals distinct relationships between lake areas and water consumption types. Halaqi Lake's area strongly negatively correlates with industrial water consumption. East Juyanhai Lake's area has a pronounced positive correlation with residential water consumption. Qingtu Lake's area negatively correlates with both agricultural irrigation and residential water consumption, with correlation coefficients surpassing 0.8. Analysis indicates increasing trends in industrial and residential water consumption in the Shule River Basin, with rates of $2.82 \times 10^6$ m$^3$/a and $1.39 \times 10^6$ m$^3$/a, respectively (Figure 8). In the Heihe River and Shiyang River basins, both industrial and residential water consumption trends are declining. Industrial water consumption is dropping at rates of $2.82 \times 10^6$ m$^3$/a in the Heihe River and $1.73 \times 10^6$ m$^3$/a in the Shiyang River. Residential water consumption is also seeing reductions, with rates of $0.16 \times 10^6$ m$^3$/a in the Heihe River and $1.02 \times 10^6$ m$^3$/a in the Shiyang River. Agricultural irrigation water consumption trends differ. In the Shule River and Shiyang River basins, there is a decline, with rates of $-41.18 \times 10^6$ m$^3$/a and $-15.91 \times 10^6$ m$^3$/a, respectively. In contrast, the Heihe River basin has seen an increase at $3.06 \times 10^6$ m$^3$/a. Rising industrial water demand in the Shule River Basin conflicts with lake water supply needs and underscores water resource imbalances. In the Heihe River Basin, decreasing residential water consumption aids terminal lake expansion. Yet, increased farmland, heightened irrigation water use, and rising residential water consumption lead to reduced water inflow to the terminal lake and exacerbate lake drying [54].

Among the the factors of forestry, animal husbandry, and fisheries water consumption, urban public water consumption, and ecological environmental water consumption, Halaqi Lake's area notably correlates with water consumption for forestry, animal husbandry, fisheries, and ecological purposes. Specifically, there is a strong positive link between the lake area and water usage for forestry, animal husbandry, and fisheries. East Juyanhai Lake's area correlates more with urban public and ecological environmental water consumption, both positively. Qingtu Lake's area correlates strongly with water consumption for forestry, animal husbandry, fisheries, and urban public use. The most significant correlation is with water consumption for forestry, as shown by a coefficient of 0.91. In the Shule River, water consumption for forestry, animal husbandry, and fisheries has decreased at rates of $3.84 \times 10^6$ m$^3$/a, and urban public water consumption has decreased at a rate of $0.52 \times 10^6$ m$^3$/a. In contrast, in the Heihe River and Shiyang River, water consumption for forestry, animal husbandry, and fisheries, as well as urban public water consumption, has shown an increasing trend; the rates of change are $3.95 \times 10^6$ m$^3$/a, $0.57 \times 10^6$ m$^3$/a, $6.37 \times 10^6$ m$^3$/a, and $0.29 \times 10^6$ m$^3$/a, respectively. Ecological environmental water consumption is increasing across all three basins. The rate of increase for ecological environmental water consumption in the Shule River is $51.64 \times 10^6$ m$^3$/a, while the rates of increase for the Heihe River and Shiyang River are $3.42 \times 10^6$ m$^3$/a and $8.31 \times 10^6$ m$^3$/a, respectively. The positive correlation between the area of Halaqi Lake and water consumption for forestry, animal husbandry, and fisheries may arise from a reduction in water consumption in the Shule River Basin. This reduction could lead to an increase in river runoff flowing into the terminal lake. Measures of water resource management that involve wetland restoration and improving water use efficiency in forestry, animal husbandry, and fisheries might have also played roles. There is a negative correlation between the lake area and ecological environmental water consumption. As ecological awareness grows, and efforts to enhance the environment intensify, ecological water consumption rises. This benefits ecosystem restoration but also strains river runoff. The Heihe River's increased flow in recent years has augmented the water supply for lakes, human activities, and ecological improvements. Hence, there is a positive correlation between urban public water consumption, ecological environmental water consumption, and lake area in this region. In the Shiyang River Basin, water used for forestry does not return to surface water bodies or groundwater aquifers. Yet, moisture-absorbing vegetation aids soil in retaining moisture, promoting precipitation infiltration and groundwater replenishment. This stability in the water cycle ensures consistent groundwater and river flow volumes.

Thus, the strongest correlation is between Qingtu Lake's area and water consumption for forestry, animal husbandry, and fisheries.

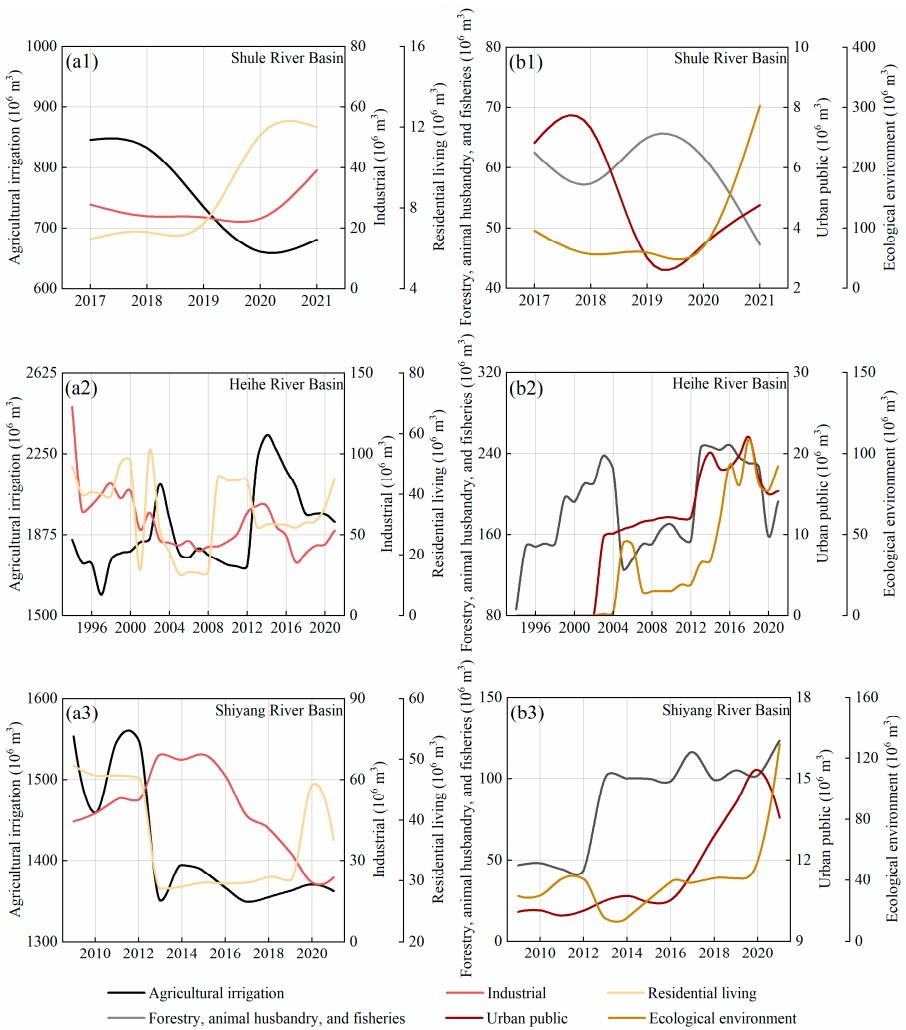

**Figure 8.** (**a1**,**b1**) Changes in water consumption in the Shule River Basin from 2017 to 2021 (including agricultural irrigation, industrial, residential living, forestry, animal husbandry, and fisheries, urban public, and ecological environment); (**a2**,**b2**) changes in water consumption in the Heihe River Basin from 1994 to 2021; and (**a3**,**b3**) changes in water consumption in the Shiyang River Basin from 2009 to 2021.

Ecological water conveyance plays a pivotal role in counteracting the impacts of human-induced water consumption on terminal lakes. In the Hexi Interior, to counterbalance the challenges posed by excessive water resource development, ecological water conveyance has emerged as a key strategy for balancing the needs of both the socioeconomic and natural systems [55]. In the Shule River Basin, the local government initiated an ecological water conveyance project in 2011 and implemented the "Comprehensive Planning for the Rational Utilization of Water Resources and Ecological Conservation in Dunhuang" to address water resource management and ecological conservation issues. As of 2019, the water volume reaching the downstream end North Lake Beach from the Danghe River was $0.42 \times 10^8$ m³, and the water volume reaching the downstream end North River Mouth from the Shule River was $1.67 \times 10^8$ m³. Concurrently, the area of Halaqi Lake reached its peak at 23.03 km². During this period, the primary source of water supply for Halaqi Lake was ecological water conveyance from the Shule River. Similarly, in the Heihe River Basin and the Shiyang River Basin, ecological water conveyance plans were jointly implemented, including the "Heihe River Mainstream Flow Allocation Plan" and the

"Shiyang River Basin Priority Remediation Plan". In the Heihe River Basin, starting from the year 2000, ecological water conveyance from the Yingluo Gorge Hydrological Station to the Zhengyi Gorge Hydrological Station and Langxinshan Hydrological Station amounted to $14.62 \times 10^8$ m$^3$ (Figure 9a), and it increased at a rate of 0.19/a thereafter. As of 2022, the runoff volume conveyed from Yingluo Gorge Hydrological Station to Zhengyi Gorge Hydrological Station and Langxinshan Hydrological Station has exceeded $400 \times 10^8$ m$^3$. Concurrently, as the ecological water volume conveyed from Yingluo Gorge Hydrological Station increased, the inflow into East Juyanhai Lake also grew. From 2002 to 2022, the inflow into the lake increased by 23%, with a total runoff volume into the lake surpassing $12 \times 10^8$ m$^3$. The ecological water conveyance project in the Shiyang River Basin began implementation in 2008. The conveyed ecological water volume flows through the Caiqi Hydrological Station into the Hongyashan Reservoir and ultimately enters Qingtu Lake. During the period from 2008 to 2022, the cumulative total conveyance volume at the Minqin Caiqi Hydrological Station exceeded $44 \times 10^8$ m$^3$ (Figure 9b), the cumulative total conveyance volume at the Hongyashan Reservoir exceeded $3 \times 10^8$ m$^3$, and the final inflow volume into Qingtu Lake exceeded $1.9 \times 10^8$ m$^3$. There is a significant positive correlation between ecological water conveyance and changes in lake area, with a correlation coefficient exceeding 0.75. The inflow volume into the lake expanded with the introduction of ecological water conveyance, leading to the restoration of the ecological state of Qingtu Lake. In September 2009, the lake surface area reached 0.09 km$^2$, and by 2022, Qingtu Lake's area had stabilized at 2.60 km$^2$. However, despite these efforts, the areas of all three lakes witnessed a decline in 2021–2022. This suggests that the current water conveyance strategies might be inadequate to sustain larger lake areas. Among human-influenced factors, ecological water conveyance shows the strongest correlation with the area of East Juyanhai Lake and a significant positive correlation with Qingtu Lake's area. In an era where human activities are intensifying lake shrinkage, ecological water conveyance stands out as an essential tool. It not only supports the ecological health of the lakes but also ensures the stability of surrounding ecosystems and the conservation of biodiversity.

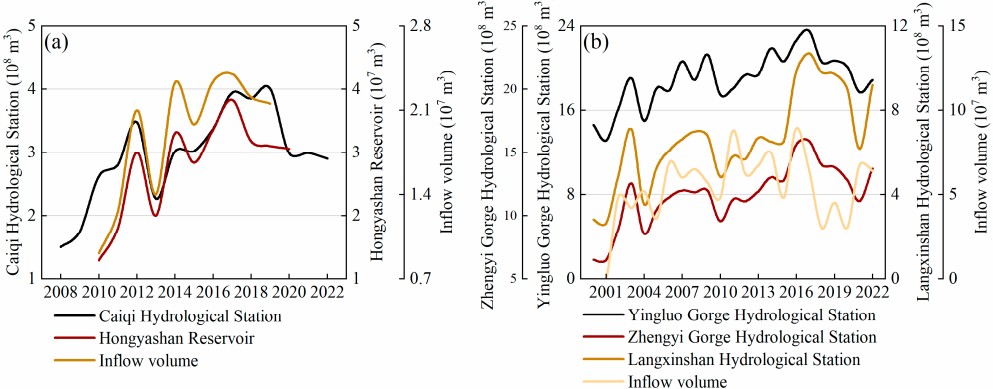

**Figure 9.** (**a**) Ecological water conveyance and inflow volume from Yingluo Gorge, Zhengyi Gorge, and Langxinshan Hydrological Stations in the Heihe River Basin; and (**b**) ecological water conveyance and inflow volume from the Caiqi Hydrological Station and Hongyashan Reservoir in the Shiyang River Basin.

In terms of the impact of human activities on Halaqi Lake, East Juyanhai Lake, and Qingtu Lake, different response patterns have emerged. The area change of East Juyanhai Lake closely links to the volume of ecological water conveyance, indicating that human water resource management and allocation have a significant impact on the size and health of the lake. Meanwhile, the changes in the area of Halaqi Lake and Qingtu Lake are primarily attributed to industrial water consumption, as well as water consumption for forestry, animal husbandry, and fisheries activities in the upper reaches of the basin. This sensitivity arises from lakes transitioning from dry to reappearing, making them particularly responsive to surrounding water use activities. Industrial water consumption along with

water for forestry, animal husbandry, and fisheries plays a direct role in determining the lakes' existence.

As natural reservoirs, terminal lakes aid in regulating regional water cycles and climates. Lakes exert a significant impact on local climate through evaporation and infiltration, assisting in mitigating extreme temperatures and arid conditions. With unique hydrological characteristics, these lakes have the capacity to store water during drought periods, serving as a lifeline for surrounding ecosystems. Terminal lakes also assume a central role in maintaining biodiversity [56], offering habitats to numerous unique species, including vegetation and terrestrial animals reliant on aquatic ecosystems. The health of these ecosystems directly links to regional biodiversity and ecological stability. Terminal lakes prove pivotal for local livelihoods, providing fishing resources, tourism potential, and cultural value [57]. Lakes play an irreplaceable role in local economic and social development. Protecting terminal lakes in ecologically fragile areas goes beyond preserving individual ecosystems—it is instrumental in safeguarding regional ecological security and promoting sustainable development. This necessitates comprehensive management measures to ensure the rational use of water resources and the long-term stability of the ecological environment.

## 6. Conclusions

Using the Google Earth Engine (GEE) platform along with Landsat TM/ETM+/OLI and Sentinel-2 MSI data, this study systematically examined annual changes in Halaqi Lake (2017–2022), East Juyanhai Lake (1990–2022), and Qingtu Lake (2009–2022) within the Hexi Interior. The study analyzed spatiotemporal dynamics in lake areas and assessed the impact of climate change and human activities on these changes. From this analysis, several main conclusions emerged:

1. Since 1990, the surface areas of Halaqi Lake, East Juyanhai Lake, and Qingtu Lake have experienced dynamic shifts such as drying, expansion, and contraction. Among them, Halaqi Lake formed a water surface area of 13.49 km$^2$ in 2017, attained its peak extent of 23.03 km$^2$ in 2019, and was subsequently reduced to 9.53 km$^2$ by 2022. East Juyanhai Lake encountered dry conditions in the years 1992, 1995, 2001, and 2002. Subsequently, the lake underwent rapid expansion, achieving its peak area of 69.09 km$^2$ in 2019. Nevertheless, by 2022, the lake's area had declined to 40.84 km$^2$. In 2009, Qingtu Lake's area expanded from 0.00 km$^2$ to 0.09 km$^2$. It reached its maximum extent in 2017 (11.63 km$^2$). However, from 2017 to 2022, the lake's area steadily decreased, with Qingtu Lake's water surface area diminishing to 2.60 km$^2$ by 2022;

2. The area of the lakes primarily results from the combined influences of climate change and human activities. East Juyanhai Lake's area is notably more affected by climate change, while Halaqi Lake and Qingtu Lake are more impacted by human activities. Among these lakes, the most prominent factors influencing area changes are industrial water consumption (0.98) for Halaqi Lake, temperature (0.6) for East Juyanhai Lake, and water consumption for forestry, animal husbandry, and fisheries (0.91) for Qingtu Lake. The varying geographical locations led to significant disparities in the correlation between climate and human activities affecting the area of these three lakes. Nonetheless, it is essential to note that the management of water resources through human activities stands as the primary cause of sudden area changes in these lakes;

3. Terminal lakes depend on residual water from upstream usage and often do not receive adequate priority. Therefore, the fundamental approach to the ecological restoration of terminal lakes is the release of ecological water. The amount of water released each year needs to be adjusted based on dynamic meteorological conditions and the lake's water budget. Different lake management goals and related environmental water requirements should be established under three different meteorological conditions: wet years, normal years, and dry years. This also involves considering the evapotranspiration of natural vegetation, the evaporation from the basin, and the

seepage from lakes and rivers. While these actions have been proven beneficial in the short term, further measures, such as improving the efficiency of agricultural irrigation, increasing the reuse of industrial water, and expanding wastewater treatment and reuse can help reduce upstream water withdrawals for production and domestic use. This will play a significant role in improving the local ecology.

**Author Contributions:** Conceptualization, Q.M. and X.Y.; methodology, Q.M.; validation, C.Z., C.Y., K.Y., Z.T. and J.L.; writing—original draft preparation, Q.M.; writing—review and editing, Q.M., X.Y. and C.Z. All authors have read and agreed to the published version of the manuscript.

**Funding:** This research was funded by the Strategic Action Plan of Oasis Science (grant number: NWNU-LZKX-202301).

**Institutional Review Board Statement:** Not applicable.

**Informed Consent Statement:** Not applicable.

**Data Availability Statement:** Data are contained within the article.

**Acknowledgments:** We are grateful to the School of Geography and Environmental Science, Northwest Normal University; the Strategic Action Plan of Oasis Science (grant number: NWNU-LZKX-202301). We thank the anonymous reviewers and editorial staff for their constructive and helpful suggestions.

**Conflicts of Interest:** Author Xiaojun Yao was employed by the University Northwest Normal University and Company Academician's Studio of Gansu Dayu Jiuzhou Space Information Technology Company Limited. The remaining authors declare that the research was conducted in the absence of any commercial or financial relationships that could be construed as a potential conflict of interest.

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
