# Peer review of "Spatio-Temporal Dynamics of Terminal Lakes in the Hexi Interior, China"

_sustainability, doi:10.3390/su16010211_

Round 1

Reviewer 1 Report

Comments and Suggestions for Authors

The manuscript sets out to explain changes in terminal lake area using diverse physical variables. remote sensing data. It takes time to read it because of the complexity of the subject matter, but it´s quite well written.

Line 88: I´m not sure “evaluate” is the best choice of a verb. Is there really an evaluation, comparing it to some standard? I don´t think so.

Lines 152-153: This part of the methodology is unclear. Can it be further explained, so that the process can be replicated?

Figures 3 and 4: Can the letters a, b and c be made larger or stand out in some way? Took me a while to find them.

Lines 305-307: This explanation related to climate change is important. Are there other references explaining the relation about glacier melting and lakes? Is this happening in other countries and continents? If not, why? These could be studies done elsewhere, not only about the Hexi interior and not only in China. Please help other researchers abroad, who may be interested in this topic. This will make the research more generalizable and useful.

Line 382. Starting on this line there are many “106” and “108” that are missing the superscript in the last digit.

Lines 487-480. Are there any references to support this? This is important, but the ideas are not fully developed yet.

Line 497: Are you sure the study monitored the changes? The word “monitor” means to observe over a period of time, not just to take historical data and analyze it.

Lines 512-513: This is important and could be better discussed in Section 5 of the manuscript. There may be references for this that should be added. These could be about other places, not just in China.

Line 523: The ending paragraph should be stronger, given the strength of the data and interpretation. Please add a paragraph with recommendations for other researchers and/or policy makers.

Comments on the Quality of English Language

English is quite good. Grammar is unclear in lines 101-103.

Author Response

Thank you for your approval of this study and for your comments, which are valuable for revising and improving our manuscript with important guiding significance. We have made corrections according to the comments, and the revised portions are marked in red in the revised manuscript. The responses to the reviewer's comments are as follows.

Reviewer 2 Report

Comments and Suggestions for Authors

This paper uses Landsat TM/ETM+/OLI 17 and Sentinel-2 MSI to study the temporal and spatial dynamics and influencing factors of terminal lakes, which has strong practical significance. However, there are still many areas that need to be improved.

Major:

1. Meaning and characteristics of terminal lake. It is recommended to explain the meaning and characteristics of the terminal lake and the reasons for choosing the terminal lake area in the introduction or study area section.

2. The Halaqi Lake appeared in 2017 and The Qingtu Lake appeared in 2009, but the title of the paper is 1990-2022. Is this contradictory?

3. The author discusses the spatio-temporal dynamic evolution process of the three terminal lakes, but due to the length of the paper, it cannot explain all the problems clearly. Therefore, I suggest that the author choose a terminal lake for in-depth analysis of the impact of climate change and human activities on her.

4. The formation reasons and dynamic change processes of the three lakes are different, and they all have their particularities. The authors did not compare the three lakes or explain why these areas were chosen for the study.

5. The authors use various types of water use to represent human activities and study their impact on terminal lakes. Is that reasonable?

Comments on the Quality of English Language

 Moderate editing of English language required

Author Response

(The authors gave the same response as above.)

Reviewer 3 Report

Comments and Suggestions for Authors

General evaluation

The manuscript title does not fit with the work performed. Despite it performed a spatial-temporal dynamics, the manuscript failed on causal analysis. They described many driven factors, but they do not explore further which driven factors are responsible for the dynamics of Terminal Lakes. Thus, the conclusions are not supported by the work.

I recommend changing the title for “spatial-temporal dynamics of Terminal Lakes in the Hexi Interior, China”, discussing the alterations observed and highlighting the importance of preservation of such ecosystems.

Minor Comments:

Figure 1: for readers outside the China, it is not possible to identify where the study area is located. I recommend the authors to change the figure, giving more details on the study area

Figure 5: I am not sure if lake shoreline length can really support a discussion, once it depends on fractal dimensions, and consequently, the measurement errors could be larger than inherent uncertainties on this topic. Also, the comparison between the boxplots among the years did not show significant differences between the periods. The exception is for Lake Area from 1990 to 2022 to East Juyanhai Lake (Fig 5b). It reinforces that the results do not support the conclusions made by the authors.

Comments on the Quality of English Language

No major concerns regarding the English Quality

Author Response

(The authors gave the same response as above.)

Round 2

Reviewer 2 Report

Comments and Suggestions for Authors

The author has revised and improved the paper, but there are still big problems to be improved before it can be published.

1. The title of the paper has been deleted (1999-2022), but this time still appears in a large number of places in the paper. Is that reasonable?

2. The authors say that water consumption can indicate the impact of human activities on lakes, but they do not see enough evidence in the paper. I suggest adding the basis of index selection in the part of method system.

3. The paper has been revised, but the abstract does not seem to have been revised. Is there a contradiction in that?

4. What are the human factors that affect the temporal and spatial evolution of water volume in these lakes? How can these factors be controlled to protect the lake ecosystem?

5. The authors prove in the abstract that climate change will cause changes in lake water volume, which seems to be meaningless. I suggest that the authors mainly analyze the effects of human activities, not climate change. Because climate change will cause glaciers to melt, which will change the amount of water in the lake. But how does human activity affect it is worth studying and discussing

Author Response

(The authors gave the same response as above.)

Reviewer 3 Report

Comments and Suggestions for Authors

The authors have improved the manuscript's quality. The manuscript is ready for publication.

Author Response

Thank you for your approval of this study and for your comments, which are valuable for revising and improving our manuscript with important guiding significance. The manuscript has been revised for sentence grammar and other issues, please refer to the revised manuscript for more details.

Round 3

Reviewer 2 Report

Comments and Suggestions for Authors

Accept in present form